# *Paramecium bursaria* as a Potential Tool for Evaluation of Microplastics Toxicity

**DOI:** 10.3390/biology11121852

**Published:** 2022-12-19

**Authors:** Jianhua Zhang, Changhong Li, Xiangrui Chen, Yunqian Li, Chenjie Fei, Jiong Chen

**Affiliations:** 1State Key Laboratory for Managing Biotic and Chemical Threats to the Quality and Safety of Agro-Products, Ningbo University, Ningbo 315832, China; 2Laboratory of Biochemistry and Molecular Biology, School of Marine Sciences, Ningbo University, Ningbo 315832, China

**Keywords:** *Paramecium bursaria*, microplastics, toxicological tool, protozoa, unicellular organisms

## Abstract

**Simple Summary:**

Various multicellular animals are currently used as tools for the toxicological assessment of tiny plastic wastes (i.e., microplastics) in aquatic systems. Here, we present *Paramecium bursaria* as a promising unicellular alternative for evaluating the toxicity of microplastics due to their easy cultivation, low cost, and fewer ethical considerations. We observed a range of behavioral and molecular changes in *Paramecium bursaria* in response to microplastic exposure. These elicited changes include abnormal swimming patterns, reduced moving speed, impaired predator avoidance, and elevated oxidative stress. Overall, we demonstrated the possibility of using *Paramecium bursaria* as an alternative eukaryotic species for evaluating the toxicity of microplastics, which may serve as a valuable tool in the study of other environmental contaminants.

**Abstract:**

Microplastics (MPs) are normally defined as small plastic wastes with a size of 1 μm to 5 mm in diameter. This tiny plastic debris is abundant in aquatic systems and poses a great threat to aquatic biota. To date, toxicological assessment of MPs is predominantly dependent on metazoan animals, although their applications are sometimes limited due to the high cost, narrow ecological niche, or ethical considerations. In this regard, unicellular eukaryotes (i.e., protozoa) that are ubiquitously present in nature represent a promising alternative for evaluating the toxicity of MPs. In this study, we selected *Paramecium bursaria* (*P. bursaria*) as a representative of protozoa and further investigated behavioral and molecular changes in MPs-exposed *P. bursaria*. Our results showed that following MPs uptake, *P. bursaria* exhibited various changes, including anomalies in swimming patterns, reduction in moving speed, impairment of avoidance behavior, elevation of oxidative stress, and potential disturbance of endosymbiosis. These elicited changes in *P. bursaria* in response to MPs exposure were pronounced and measurable. Overall, this study demonstrated that *P. bursaria* could serve as a promising alternative for the toxicological assessment of MPs and may be further applied to evaluate the toxicity of other environmental contaminants.

## 1. Introduction

Plastic contamination is a major environmental problem in aquatic systems, and its adverse impacts on local biota are primarily determined by the size of the debris [1]. Large plastic debris, such as discarded fishing lines and bottle caps, cause physical entanglement and intestinal blockage among various aquatic organisms [2,3]. Further, these large plastic wastes can be fragmented into microplastics (MPs) through physical abrasion, photo-oxidation, and biodegradation [4]. The small size of MPs, which normally range from 1 μm to 5 mm in diameter, makes them more accessible to various aquatic species, likely through direct ingestion or trophic transfer, and consequently, represent an emerging threat to the aquatic biota [5].

To date, numerous efforts have been devoted to assessing the potential toxicity of MPs. Since the concentration of this novel contaminant in the aquatic system remains elusive, various concentrations, ranging from 10^2^ to 10^10^ particles/mL, were selected to evaluate acute or chronic impacts on aquatic animals from different trophic positions under experimental settings [6,7,8]. Exposure to MPs has led to changes at multiple levels in examined organisms, such as reduced locomotor activities in mysid shrimp (*Neomysis japonica*), limited predator avoidance in hermit crabs (*Pagurus bernhardus*), altered metabolic functions in zebrafish (*Danio rerio*), and elevated oxidative stress in Monogonont Rotifer (*Brachionus koreanus*) [9,10,11,12]. Of note, our knowledge regarding MPs is predominantly derived from studies using multicellular animals (i.e., metazoans). However, elicited changes in response to MPs exposure in unicellular animals (i.e., protozoans) remain largely unknown. Further, the potential applications of using protozoans as alternative tools for evaluating MP toxicity are currently underappreciated.

Protozoans represent an assortment of single-celled organisms that are ubiquitously found in nature. Among these microscopic animals, *Paramecium* is of particular interest to the scientific community and has been used as a valuable model organism in eukaryotic biology and biomedical research [13,14]. In general, *Paramecium* is a genus of ciliates with short cilia arranged in rows on the membrane and represents the first ciliates to be observed due to their relatively large size [15]. Since their discovery, numerous culturing protocols have been established, and overall, it is fairly easy to grow and maintain *Paramecium* cultures with minimal effort, relatively low cost, and fewer ethical considerations [16]. Additionally, *Paramecium* was later demonstrated to exhibit complex and yet measurable behaviors, such as prey pursuit and avoidance reactions, in response to extrinsic stimulus [17,18]. Further, genome information of several *Paramecium* species is currently available due to advances in sequencing technology, and this further offers a possibility to link changes in phenotypes to genotypes [19]. Collectively, these features make *Paramecium* excellent model organisms with the potential to assess the toxicity of MPs. Recently, a few studies have demonstrated the uptake of MPs in *Paramecium* [20,21]. However, potential impacts on *Paramecium*, especially elicited changes at behavioral and molecular levels following MPs uptake, remain to be investigated.

In the present study, *Paramecium bursaria* (*P. bursaria*) was selected as a representative species of *Paramecium*, which normally established endosymbiosis with green algae [22]. This provides an extra dimension of information regarding the impacts of MPs on endosymbionts and could potentially serve as a sensitive indicator of toxic effects. Following the uptake of MPs, locomotor activities, prey pursuit and avoidance behaviors, oxidative stress status, and the endosymbionts content of *P. bursaria* were monitored. Our results showed elicited changes at behavioral and molecular levels in MPs-exposed *P. bursaria.* Overall, this study demonstrates the possibility of using *P. bursaria* as an alternative eukaryotic species for evaluating the toxicity of MPs and will serve as a valuable tool for the toxicological assessment of other environmental contaminants.

## 2. Materials and Methods

### 2.1. Cell Culture

*P. bursaria* were cultured in E3 medium (0.29 g/L NaCl, 13 mg/L KCl, 44 mg/L CaCl_2_, 81 mg/L MgSO_4_, 0.48 g/L HEPES, pH 7.0) supplemented with *Escherichia coli* TG1 (OD_600_ = 1.0) at room temperature (24 ± 1 °C) under 12:12 light–dark period [23]. To determine the growth curve, the initial density of 60 cells/mL was seeded in a 50 mL conical flask and cultured as detailed above. In the following days, the number of *P. bursaria* was manually counted under a stereoscopic microscope (OLYMPUS SZX7, Olympus Corporation, Japan) and monitored daily for two weeks. *P. bursaria* used in this study were all collected at the logarithmic growth phase.

### 2.2. Microplastic Characterizations

Yellow-green polystyrene MPs of 1 μm in diameter (Fluoresbrite™ Carboxy YG microspheres, Polysciences, Warrington, PA, USA) were purchased for this study. Prior to experiments, polystyrene MPs were washed in E3 medium three times, and then dynamic light scattering (Zetasizer Ver. 7.12, Malvern Instruments, Malvern, UK) was used to analyze the hydrodynamic meter, polydispersity index (PDI), and zeta potential of polystyrene MPs at the concentration of 1 × 10^8^ particles/mL diluted in E3 medium. MPs used in this study were all diluted in E3 medium unless detailed otherwise.

### 2.3. Toxic Effects of MPs Exposure to P. bursaria

An acute exposure experiment was performed based on previous studies with minor modifications [24]. Briefly, *P. bursaria* were randomly distributed into a 96-well plate (10 cells/well) before being subjected to E3 medium (i.e., control group) and various concentrations of MPs ranging from 1 × 10^2^ to 1 × 10^9^ particles/mL at a 10-fold increase in the concentration. After 24 h exposure, the number of viable *P. bursaria* was manually counted under a stereoscopic microscope, and cells that lacked movement were considered dead and excluded from the analysis.

### 2.4. Flow Cytometric and Fluorescent Microscopic Analysis of MPs Uptake and Accumulation in P. bursaria

*P. bursaria* were seeded into a 24-well plate (~500 cells/well) and subjected to E3 medium alone or MPs at high (i.e., 1 × 10^8^ particles/mL), moderate (i.e., 1 × 10^5^ particles/mL), and low (i.e., 1 × 10^2^ particles/mL) concentrations, respectively. After 24 h exposure, *P. bursaria* were centrifugated at 300 × g for 10 min, and the cells were fixed using 4% buffered Paraformaldehyde (PFA). After three washes in PBS, aliquots of fixed cells were analyzed for MPs uptake using MACSQuant Analyzer 10 flow cytometry (Miltenyi Biotec, Bergisch Gladbach, Germany), and for each sample, cells from four technical replicates were pooled. At least 1000 events were collected, and data were analyzed using FlowJo™ v10.8 Software (BD Life Sciences, Franklin Lakes, NJ, USA). For fluorescent microscopy, ~20 μL of fixed cells was mounted on microscopic slides and imaged using a fluorescence microscope (Nikon, Tokyo, Japan). To further locate internalized MPs, *P. bursaria* exposed to MPs (i.e., 1 × 10^8^ particles/mL) were stained using 0.1% Congo red dye (Solarbio, Beijing, China) diluted in PBS for 1 h to label food vacuoles [25] prior to fixation and imaging as detailed above.

### 2.5. Effects of MPs Exposure on P. bursaria Locomotion Activity, Avoidance Behaviors, Prey Pursuit, and Predator Evasion Ability

*P. bursaria* were seeded in a 24 well-plate and then exposed to E3 medium alone or MPs at concentrations of 1 × 10^2^, 1 × 10^5^, and 1 × 10^8^ particles/mL, respectively. After 24 h treatment, individual *P. bursaria* were randomly picked and transferred to a droplet of E3 medium (~5 µL) on a slide (Citotest Scientific Corporation, Haimen, China), and then trajectories of individual *P. bursaria* were monitored and recorded for 2 min (2 frames per second) using microscopy (ECLIPSE Ni-E, Nikon, Tokyo, Japan). Moving trajectories were further plotted and analyzed to obtain mean/maximum speed using TrackMate plug-in in the Image J (Fiji) program [26]. All experiments were performed in sextuplicate.

To investigate the impacts of MPs exposure on prey pursuit and avoidance behaviors of *P. bursaria*, a two-well testing apparatus was customized. Briefly, this testing apparatus has two wells (designated as “well A” and “well B”; ~16 millimeters (mm) in diameter and 11.5 mm in height) that are connected by a canal (designated as “C”; ~20 mm in length). Two dividers can be inserted at both ends of the canal to form three enclosed spaces. This two-well testing apparatus was designed using Shapr3D software (Ver. 5.91) and printed in resin with a 3D printer (SLA600, JGMAKER, Shenzhen, China). For the prey pursuit assay, *P. bursaria* were treated the same as detailed above, and on the day of experiments, the testing apparatus was filled with E3 medium, followed by the insertion of dividers. Then, TG1 (OD_600_~0.5), serving as attractants, were added to well A, after which ten *P. bursaria* were transferred to the middle region of the canal prior to the careful removal of two dividers. *P. bursaria* were then allowed to travel freely for 10 min, and then the number of cells in well A and well B was counted. To test avoidance behaviors, the same experimental set-up was followed as detailed above, with the exception that 2.5% glycerol, which is lethal to *P. bursaria*, was added in well A and served as a repellent. All experiments were performed in triplicate.

To examine predator evasion ability, *P. bursaria* were treated the same as detailed above, and on the day of the experiment, ten *P. bursaria* were randomly picked and co-incubated with two *Cyclops* (Copepoda: Cyclopoida), which are considered a natural predator of *P. bursaria* [27]. The number of surviving *P. bursaria* was counted every 15 min until all *P. bursaria* in any group were predated. All experiments were performed in triplicate.

### 2.6. Examination of Oxidative Stress Status of P. bursaria following MPs Exposure

*P. bursaria* were exposed to E3 medium alone or MPs at concentrations of 1 × 10^2^, 1 × 10^5^, and 1 × 10^8^ particles/mL, respectively. After 24 h exposure, *P. bursaria* were centrifugated at 800× *g* for 15 min, and the cell pellet (~2500 cells) was resuspended with 5 mL of pre-chilled PBS prior to ultrasonication at 4 °C; lysates were centrifuged at 13,000× *g* for 20 min at 4 °C, and then the supernatant (1 mL) was collected to analyze activities of two canonical antioxidant enzymes; specifically, superoxide dismutase (SOD) and catalase (CAT), using SOD and CAT assay kit (Jiancheng Biotechnology Research Institute, Nanjing, China) according to manufacturers’ instructions.

To quantify the contents of photosynthetic pigments, including Chlorophyll a (Chl-a) and Carotenoids (Car), *P. bursaria* were treated as detailed above, and pelleted cells (~25,000 cells) were resuspended in 5 mL of pre-chilled methanol. After ultrasonication, the organic fraction was collected and left in the fridge overnight. The next day, the organic fraction was centrifugated at 13,000× *g* for 20 min at 4 °C, and supernatant (1 mL) was collected for measuring optical density (OD) values at 480, 510, 652, 665, and 750 nm using a UV spectrophotometer (METASH model 6100, Shanghai, China). The content of Chl-a and Car was calculated using the following equations: Chl-a (μg/mL) = 16.29 × (OD_665_ − OD_750_) − 8.54 × (OD_652_ − OD_750_); Car (μg/mL) = 7.6 × (OD_480_ − OD_750_) − 1.49 × (OD_510_ − OD_750_), as detailed in [28].

### 2.7. Statistics

The ordinary one-way ANOVA test was used for the statistical evaluation of all the results. Values are shown for data that reached a significance of *p* ≤ 0.05 (∗), *p* ≤ 0.01 (∗∗). All data is shown as the mean ± SD, and all statistical analyses were performed using GraphPad Prism Ver. 9.4.1 (GraphPad Software, San Diego, CA, USA). The survival analysis was performed using the Kaplan–Meier method (* *p* < 0.05).

## 3. Results

### 3.1. Physicochemical Characterization of MPs and Determination of Toxic Concentration of MPs on P. bursaria Growth

The size distribution of MPs was 1036 ± 24 nm on DLS, which is fairly consistent with the size advertised by the manufacturer. DLS analysis further revealed that MPs in E3 medium were highly stable with a zeta potential of 38 ± 3 mV and exhibited a very low aggregation rate, as demonstrated by the low polydispersity index of 0.05 ± 0.016 (Table 1).

The growth curve of *P. bursaria* was first determined, and the result showed a characteristic sigmoidal growth pattern that includes a lag phase (day 0 to day 3), a logarithmic growth phase (day 4 to day 9), and a plateau phase from day 10 to day 15 (Figure 1A).

To further investigate the impact of MPs exposure on *P. bursaria* proliferation, cells at the logarithmic growth phase were collected and exposed to MPs at various concentrations. The result showed that significant inhibition on cell proliferation was only observed when *P. bursaria* were exposed to the highest concentration (i.e., 1 × 10^9^ particles/mL) of MPs; comparatively, although cell counts were slightly decreased in cells exposed to MPs ranging from 1 × 10^2^ to 1 × 10^8^ particles/mL, no significant difference was observed compared to the control group (Figure 1B). Accordingly, three concentrations, specifically, 1 × 10^8^, 1 × 10^5^, and 1 × 10^2^ particles/mL that showed no inhibitory effect on *P. bursaria* growth, were selected and designated as high, moderate, and low concentrations, respectively, and used in all experiments detailed below.

### 3.2. Accumulation of MPs in Food Vacuoles

To investigate the uptake of MPs, *P. bursaria* were exposed to fluorescent MPs at high (i.e., 1 × 10^8^ particles/mL), moderate (i.e., 1 × 10^5^ particles/mL), and low concentrations (i.e., 1 × 10^2^ particles/mL), respectively, and flow cytometric analysis demonstrated the ingestion of MPs by *P. bursaria* as percentages of fluorescent cells increased to 99.1% (exposed to MPs at high concentration) and 82.8% (exposed to MPs at moderate concentration), respectively; this represents a pronounced increase in percentages of positive cells compared to cells exposed to MPs at low concentration or untreated cells (Figure 2A). Further fluorescent intensity analysis demonstrated that a significant increase in median fluorescent intensity (MFI) was observed in cells exposed to MPs at high (i.e., MFI = 428.67 ± 132.68) concentration in comparison to the control group (i.e., MFI = 0.06 ± 0.01). Comparatively, cells exposed to the low or moderate concentration of MPs exhibited slight increases in MFI, but no significant difference was found compared to the control group (Figure 2B). To further visualize the location of ingested MPs, fluorescent microscopy was performed, and consistent with the observations above, no fluorescent signal was detected in *P. bursaria* exposed to MPs of low concentration (Figure 2C). Comparatively, a punctate staining pattern was observed in *P. bursaria* as the concentration of exposed MPs increased, and when cells were exposed to the high concentration of MPs, intense signals were seen within food vacuoles in addition to punctate signals in the cytoplasma (Figure 2D).

### 3.3. MPs-Exposed P. bursaria Damage on Locomotion Activity, Prey Pursuit, and Avoidance Behaviors

To investigate the impacts of MPs exposure on *P. bursaria* locomotion activity, moving trajectories of MPs-exposed cells were recorded, and instantaneous max/mean swimming speed derived from recorded videos was further analyzed. Our results demonstrated a marked disturbance in locomotion behavior following exposure to MPs at moderate and high concentrations. Specifically, cells exposed to MPs of low concentration exhibited a basal level of locomotion activity similar to the control group. However, cells exposed to MPs at a moderate concentration were less locomotive as fewer territories were explored, and when *P. bursaria* were exposed to a high concentration of MPs, a pronounced change in locomotion activity was observed as cells exhibited an abnormal circling behavior (Figure 3A). Quantitative analysis of mean swimming speed further supported the observations detailed above, and decreases in mean swimming speed were observed in cells exposed to MPs at moderate (i.e., ~1.28-fold decrease) and high (i.e., ~1.59-fold decrease and significantly lower than the control group) concentrations (Figure 3B). Although no significant difference was found among all groups regarding the instantaneous max speed, a decrease (i.e., ~1.35 fold) in this value was seen in cells exposed to MPs at the high concentration compared to the control group (Figure 3C).

To evaluate sensory functions, a two-well testing apparatus was customized to examine prey pursuit and avoidance behavior of MPs-exposed *P. bursaria* (Figure 3D). In this study, TG1, which was the food source of *P. bursaria,* was selected as an attractant for the prey pursuit assay, and glycerol, which is lethal to *P. bursaria* at a concentration as low as 2.5% (v/v; data not shown), was selected as a repellent to test avoidance behavior. To first validate this testing apparatus, untreated cells were placed in the canal that connects well A and B, and attractant, repellent, or E3 medium alone (i.e., control group) were added to well A, respectively. The number of cells in each well was then counted after allowing cells to travel freely for 10 min. The results showed a marked prey pursuit and avoidance behavior as demonstrated by the biased distribution of cells in each well in comparison to the control group (Appendix A). To further evaluate the impacts of MPs exposure, the same experimental set-up was followed, except that cells were treated with various concentrations of MPs. In the prey pursuit assay, the number of cells in well A in which attractants were placed was consistently increased to ~10 cells among all groups, indicating that MP exposure has minimal effects on the prey pursuit. Comparatively, an increase in the number of cells in well A in which repellents were placed was observed in MPs-exposed groups. Significant differences were noted in cells exposed to MPs of moderate (~1.5-fold increase) or high concentration (~2.4-fold increase), respectively, in comparison to the control group (Figure 3E). This indicates a reduced sensory ability to avoid hazards in the surrounding environment.

To further assess the predator evasion ability following MPs exposure, MPs-exposed *P. bursaria* were co-cultured with their natural predator (i.e., *cyclops*), and the survival rate was monitored periodically. An overall decrease in the chance of survival was pronounced in MPs-exposed cells at all tested time points, which correlated with the increase in the concentration of exposed MPs. After co-culturing for 45 min, *P. bursaria* exposed to the high concentration of MPs were all predated by *cyclops,* and this represents a significant reduction in the survival rate compared to the control group (i.e., ~40%). Although no significant difference was found, a decrease in survival rates of cells exposed to MPs at moderate (i.e., ~20%) and low (i.e., ~10%) concentrations was still noted (Figure 4).

### 3.4. Impacts of MPs Exposure on the Oxidative Stress Status and Endosymbiont of P. bursaria

To assess oxidative stress status following MPs exposure, activities of two canonical antioxidant enzymes (i.e., SOD and CAT) were quantified in MPs-exposed *P. bursaria*. The results demonstrated an overall increase in the activities of both enzymes, which was positively correlated with the concentration of exposed MPs. For example, compared to the control group, a significant increase in SOD activities was observed when cells were exposed to MPs at moderate (i.e., ~4-fold increase) and high (i.e., ~6-fold increase) concentrations, respectively (Figure 5A). Although activities of CAT were significantly enhanced only in cells exposed to MPs of high concentration, an overall increase in enzyme activities was noted compared to the control group (Figure 5B).

To further investigate the impacts of MPs exposure on endosymbiotic algae, the content of two algae-derived pigments (i.e., Chl-a and Car) was quantified. As shown in Figure 5C and 5D, the content of the two pigments remained relatively consistent, and no significant difference was found among all groups.

## 4. Discussion

MPs represent an emerging threat to the aquatic environment, and numerous efforts have been undertaken to understand their distribution, transporter pathways, and potential impacts on aquatic life [29,30,31,32]. For example, MPs could absorb and enrich toxic chemicals, induce behavioral alterations, cause intestinal blockage, and interfere with the growth and reproduction in *Daphnia magna* and *Danio rerio* [33,34,35]. Although the toxicity of MPs has been tested in a range of aquatic species, their impacts on unicellular eukaryotes (e.g., *P. bursaria*) remain largely unknown. In the present study, we used *P. bursaria* as a representative of unicellular eukaryotes, and our results have shown elicited changes in both behaviors and oxidative stress status in *P. bursaria* following MPs exposure. It is important to qualify that we are not intended to understand the adverse impacts of MPs at environmentally relevant concentrations, but instead, the overall goal of this study is to investigate if changes at behavioral and molecular levels could be elicited and observed in exposed *P. bursaria*. This presents a platform that can be further customized to understand the toxicity of MPs per se or MPs associated with other contaminants. Combined with easy cultivation, fewer ethical considerations, and the relatively low cost of *P. bursaria*, we believe that unicellular organisms such as *P. bursaria* could serve as a valuable tool for future toxicity studies.

Due to their small size, MPs may potentially be ingested by plankton as they overlap with the size range of their natural prey [29]. In the present study, MPs of 1 μm in diameter were selected, as *P. bursaria* naturally prey on bacteria that normally occupy a size range from nm to µm levels [36]. Our results demonstrated the ingestion of MPs by *P. bursaria* only following exposure to MPs at moderate and high concentrations. The lack of fluorescent signals in cells exposed to MPs at low concentrations suggests an active food selection by *P. bursaria* that innately discriminates between MPs and natural prey. Although detailed mechanisms remain to be investigated, this is likely achieved through the aid of mechano- and chemoreceptors, as shown in other plankton species [37,38,39]. However, this built-in mechanism fails to prevent *P. bursaria* from uptaking MPs as the concentration of MPs increases, and consequently, food vacuoles containing aggregated MPs were observed in *P. bursaria*. Interestingly, punctate staining was also observed and possibly represented MPs translocated from food vacuoles to cortex and trichocysts via a similar route as endosymbiotic zoochlorellae [40,41]. Of note, we can not rule out the possibility that punctate signals are derived from surface-bound MPs due to the nature of this imaging technique, and further investigations are required to precisely resolve the position of MPs.

For many species, changes in behaviors often occur in response to environmental stressors [42]. These intentional or unintentional behavioral alterations impact the fitness of individuals in the changing environment. One particular behavioral change in aquatic species that is commonly investigated is locomotor activity, as changes in this behavior are directly linked to their survival [43]. Numerous studies demonstrated that following MPs exposure, changes in the locomotion of animals, including anomalies in swimming patterns and reductions in moving speed and distance, have been reported [44,45]. Consistent with these observations, *P. bursaria* that took up MPs exhibited abnormal swimming behaviors as pronounced circular trajectories accompanied by reduced moving speed were noted. Although underlying mechanisms remain to be investigated, physical disruption of membrane integrity due to MPs exposure has been documented [46]. Consequently, physical damage of the locomotory organ of *P. bursaria*, i.e., cilia, might account for the impairment of their locomotor activities; however, electron microscopy is further required. Alternatively, reduced activities of exposed *P. bursaria* are likely due to the overload of MPs, which renders them less active and lead to this abnormal swimming pattern.

Chemotaxis is a directional movement of individuals in response to chemical cues in the environment [47]. This is primarily achieved via chemoreceptors and represents one particularly important behavior for aquatic species that significantly determine their chance of survival [48,49]. Similar to other aquatic species, *Paramecium* exhibited pronounced chemotaxis, and this is likely one of the behaviors they rely on to avoid hazards and pursue prey [50]. Numerous studies demonstrated the attraction and repulsion behavior of *Paramecium* in response to a set of defined chemical cues, strongly indicating the presence of specific chemoreceptors [25,51,52]. Glycerol is a commonly used reagent to disrupt the electrical properties and integrity of membranes while maintaining a functional contractile system in *Paramecium* [53]. Consequently, *P. bursaria* would naturally exhibit avoidance behaviors in response to this chemical stimulant. However, we noted that compared to the control group, MPs-exposed *P. bursaria* were less likely to be repelled from the glycerol solution. This is likely due to the reduced locomotor activities following MP exposure, which prohibited *P. bursaria* from swiftly repelling away from this toxic chemical. Glycerol exposure would permeabilize the membrane and, thus, disrupt ion channels that are responsible for controlling cilia to propel *P. bursaria* forward or backward [54]. This loss of directionality combined with reduced locomotor activities collectively contributed to this observation that MPs-exposed *P. bursaria* are less likely to escape from this toxic chemical. Of note, there is one limitation in this experiment; glycerol is quite effective in paralyzing *P. bursaria,* and a subtle reduction in moving speed would jeopardize their survival and, thus, fail to observe the effects of MPs on the avoidance behavior in a more dynamic range, and consequently, less potent repellents should be considered in future studies. Interestingly, the ability of *P. bursaria* to locate their prey was not significantly impacted by MPs exposure. It is possible that the chemotactic behaviors of *P. bursaria* remain functional, and reduced locomotor activities due to MPs exposure have minimal effects on locating food.

Antioxidant responses have been commonly used to evaluate the impacts of various pollutants on aquatic species and serve as a sensitive indicator of oxidative stress [55]. Numerous studies demonstrated induced expressions of antioxidant enzymes following MPs exposure [56,57,58]. Similar to these observations, enhanced activities of two classical antioxidant enzymes, i.e., SOD and CAT, were observed in MPs-exposed *P. bursaria*. Organisms decrease the damaging effects of reactive oxygen species (ROS) through enzymatic antioxidant defenses. SOD and CAT both participate in antioxidant defenses, in which SOD transforms O_2_^−^ into different oxygen-derived molecular species, and catalase (CAT) degrades H_2_O_2_ to O_2_ and, collectively, mitigate the adverse effects of O_2_^−^. That being said, activities of SOD and CAT are not directly equivalent to oxidative stress level but still extensively serve as a proxy of oxidative stress status [59]. This observation further supports the notion that MPs are an oxidative stressor and likely result in an altered oxidative status of *P. bursaria* in favor of free radical generations.

Similar to other species, *P. bursaria* have the capacity to establish an endosymbiotic relationship with green algae of various strains [60]. This mutualistic endosymbiosis might represent an evolutionary adaptation to the changing environment as a way of increasing the fitness of both hosts and endosymbionts [61,62]. In this scenario, enveloped green algae under the cell cortex of *P. bursaria* are normally protected from viral infections [63]. In return, green algae could supply the nutrition and oxygen required for physiological activities (e.g., growth and respiration) of *P. bursaria* through photosynthesis [64,65]. To date, very few studies have investigated the effects of MPs on endosymbiosis, and thus in the present study, we focused on two green algae-derived pigments (i.e., Chlorophyll-a and Car) that are responsible for photosynthesis and used them as surrogates to evaluate the content of endosymbiotic algae following MPs exposure. Although no significant difference was found among all groups, slight reductions in pigment contents were observed after exposure to MPs at low and moderate concentrations in comparison to the control group. Interestingly, this trend was reversed, and a slight increase in pigment contents was seen when cells were exposed to MPs of high concentration. This is likely due to the fact that green algae were released from their hosts, as free algae were observed in medium under this condition (Data not shown). This would likely result in the uncontrolled proliferation of green algae and, thus, increase pigment contents.

## 5. Conclusions

This study represents the first report investigating behavioral and molecular changes in MPs-exposed *P. bursaria.* Reduced locomotor activities, impaired avoidance behavior, elevated oxidative stress, and potentially disturbed symbiotic relationships were observed in impacted *P. bursaria*. Overall, this study has shown elicited changes in *P. bursaria* at behavioral and molecular levels in response to MPs exposure and further demonstrates that *P. bursaria* could be a valuable tool for understanding toxicity of emerging pollutants, including MPs, in the aquatic environment or under experimental settings.

## Figures and Tables

**Figure 1 biology-11-01852-f001:**
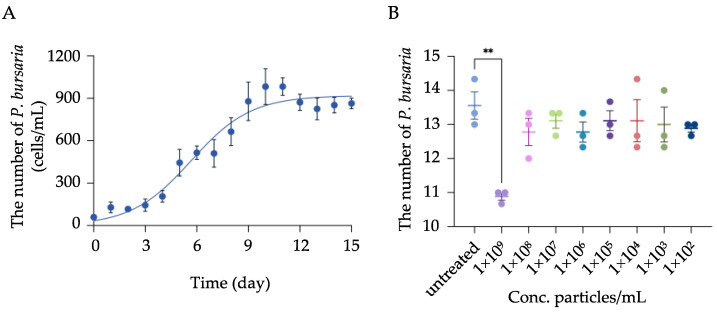
Determination of toxic concentration of MPs on *P. bursaria* proliferation. (**A**) To determine the growth kinetics of *P. bursaria*, cells were initially seeded at the density of 60 cells/mL (Day 0) in E3 medium supplemented with TG1 as a food source and cultured under a 12:12 light–dark period. Cells were manually counted every 24 h for 15 days, after which the growth curve was plotted and fitted using nonlinear regression analysis. Each dot represents the mean ± SD of three independent experiments. (**B**) Evaluation of toxic effects on *P. bursaria* proliferation following MPs exposure. *P. bursaria* (10 cells/group) at the logarithmic growth phase were collected and subjected to various concentrations of MPs ranging from 1 × 10^2^ to 1 × 10^9^ particles/mL. After 24 h exposure, viable *P. bursaria* were manually counted and compared to the control group (i.e., untreated group). The data are represented as the mean ± SD of three independent experiments, and asterisks above the line denote a significant difference (** *p* < 0.01) between means identified by the line.

**Figure 2 biology-11-01852-f002:**
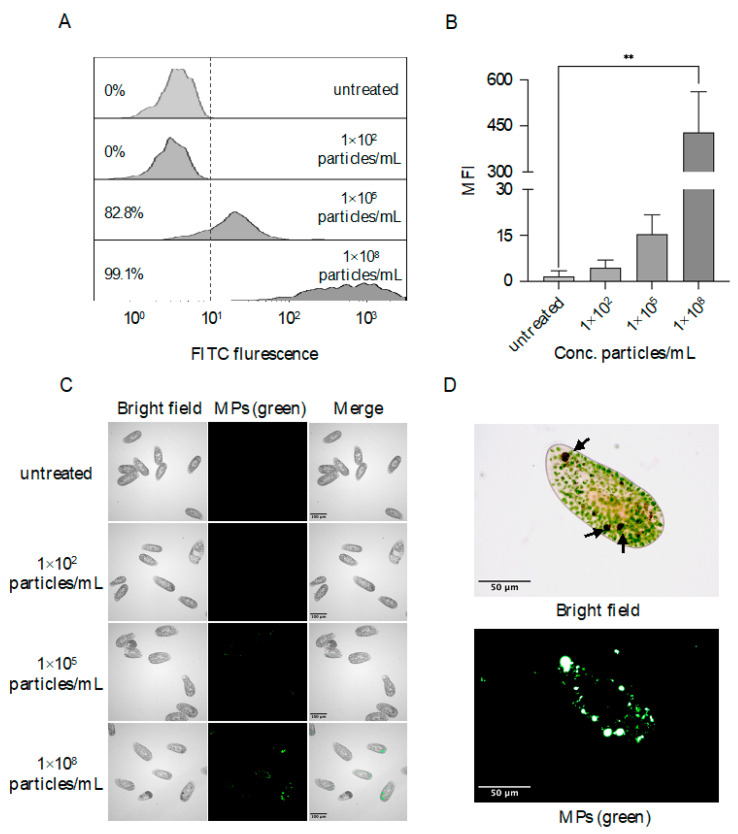
Flow cytometric and fluorescent microscopic analysis of MPs uptake by *P. bursaria*. Cells at the logarithmic growth phase were exposed to MPs at concentrations of 1 × 10^2^, 1 × 10^5^, and 1 × 10^8^ particles/mL, respectively. After 24 h exposure, cells were fixed using 4% buffered Paraformaldehyde (PFA) prior to flow cytometric (**A**) and fluorescent microscopic (**C**) analysis. At least 1000 events were collected for flow cytometric analysis, and the median fluorescent intensity of each group was calculated. The data are represented as the mean ± SD of three independent experiments, and asterisks above the line denote a significant difference (** *p* < 0.01) between means identified by the line in (**B**). A proportion of cells treated with 1 × 10^8^ particles/mL of MPs was also stained with 0.1% Congo red dye for 1 h to locate food vacuoles prior to fixation and fluorescent microscopy imaging as detailed above. Food vacuoles within *P. bursaria* are indicated by black arrows in (**D**).

**Figure 3 biology-11-01852-f003:**
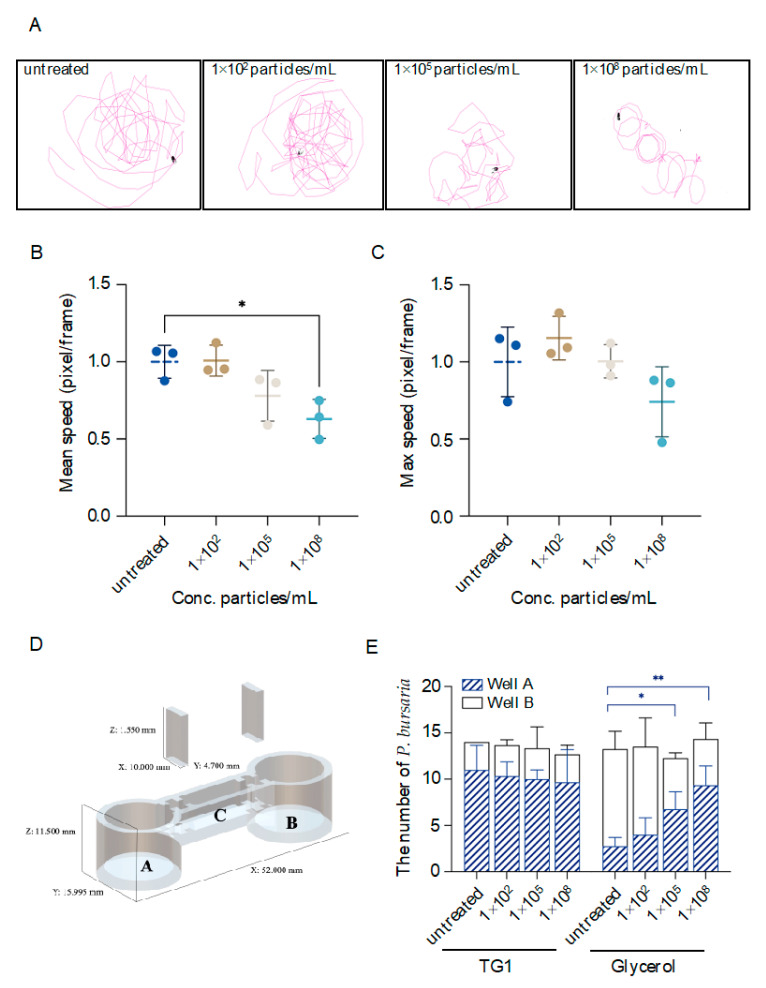
Quantitative analysis of locomotion activity, prey pursuit, and avoidance behaviors of MPs-exposed *P. bursaria*. (**A**) *P. bursaria* were exposed to MPs at concentrations of 1 × 10^2^, 1 × 10^5^, and 1 × 10^8^ particles/mL, respectively, for 24 h. On the day of the experiment, individual cells in each group were randomly picked, and their trajectories were monitored and recorded for 2 min under a microscope. Moving trajectories were further plotted and analyzed using the TrackMate plug-in in the Image J (Fiji) program to obtain mean speed and max speed, which are represented as fold change to the untreated group in (**B**,**C**), respectively. The data are represented as the mean ± SD of three independent experiments, and asterisks above the line denote a significant difference (* *p* < 0.05) between means identified by the line. (**D**) A schematic of a two-well testing apparatus for investigating prey pursuit and avoidance behaviors of *P. bursaria* following MPs exposure. This testing apparatus has two wells (designated as “A” and “B”) connected by a canal (designated as “C”). Two dividers can be inserted on both ends of the canal to form three enclosed spaces filled with E3 medium. (**E**) TG1 (food source of *P. bursaria* and served as attractants) or 2.5% glycerol (toxic to *P. bursaria* and served as repellents were added to well A and *P. bursaria* that were untreated or exposed to MPs of various concentrations were placed in canal C, respectively, and then dividers were carefully removed to allow free movement of *P. bursaria*. After 10 min, the number of *P. bursaria* in well A and B was counted and summarized in a stacked bar graph. The data are represented as the mean ± SD of three independent experiments, and asterisks above the line denote a significant difference (* *p* < 0.05, ** *p* < 0.01) between means (i.e., mean number of cells in well A) identified by the line.

**Figure 4 biology-11-01852-f004:**
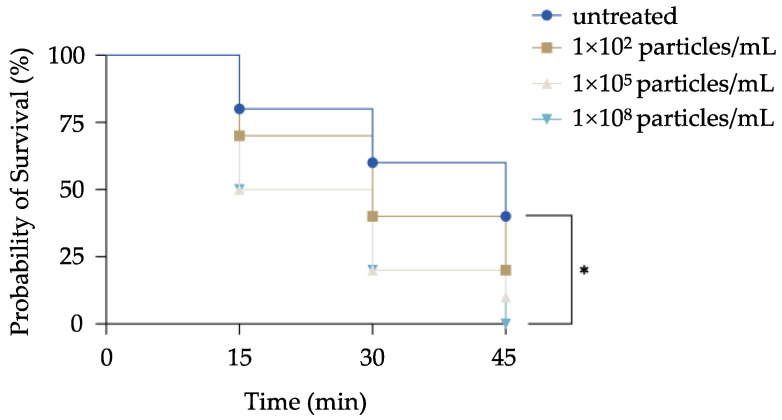
Impacts of MPs exposure on predator evasion by *P. bursaria*. *P. bursaria* were exposed to MPs at concentrations of 1 × 10^2^, 1 × 10^5^, and 1 × 10^8^ particles/mL, respectively. After 24 h exposure, ten *P. bursaria* were randomly selected and transferred into a well containing two *cyclops*. The number of survived *P. bursaria* was counted every 15 min for an hour and then plotted in a survival curve. The data are represented as the mean ± SD of three independent experiments, and asterisks above the line denote a significant difference (* *p* < 0.05) between means identified by the line.

**Figure 5 biology-11-01852-f005:**
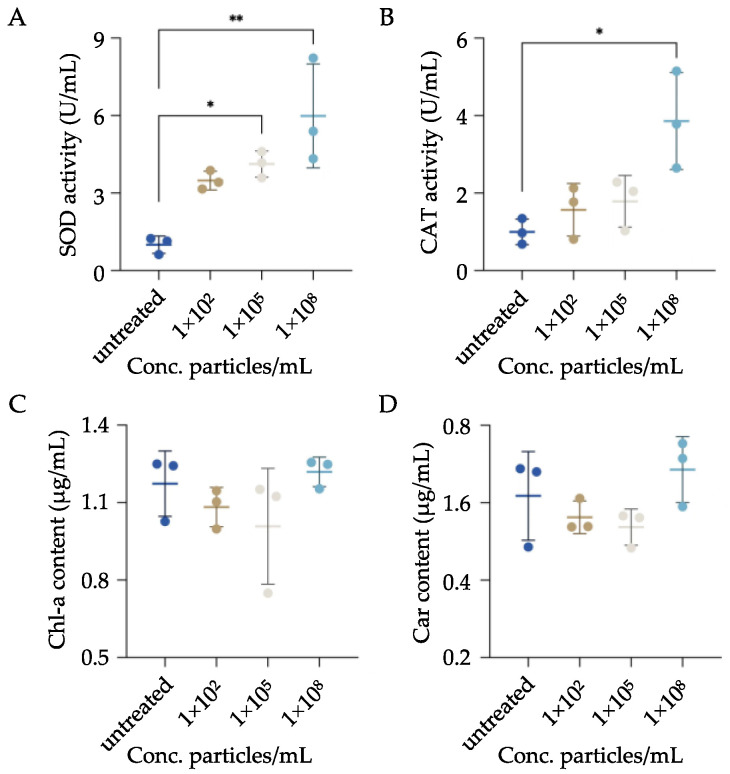
Analysis of oxidative stress status and endosymbiont contents in *P. bursaria* following MPs exposure. *P. bursaria* were exposed to various concentrations of MPs, and after 24 h exposure, cells were collected and processed as detailed in the Methods and Materials section to quantify the activities of antioxidant enzymes. Activities of superoxide dismutase (SOD) and catalase (CAT) of MPs-exposed *P. bursaria* are represented as fold change relative to the untreated group and summarized in (**A**,**B**). Contents of two pigments, chlorophyll-a (Chl-a) and carotenoids (Car), are summarized in (**C**,**D**). The data are represented as the mean ± SD of three independent experiments, and asterisks above the line denote significantly different (* *p* < 0.05, ** *p* < 0.01) between means identified by the line.

**Table 1 biology-11-01852-t001:** Physiochemical characterization of MPs.

Particles	Size, nm	Zeta-Potential, mV	PDI
Polystyrene microspheres	1003 ± 24	38 ± 3	0.05 ± 0.016

Polydispersity index (PDI) and zeta potential values were measured using DLS, and data are represented as the mean ± SD of three independent tests.

## Data Availability

The data obtained in this study are available from corresponding authors upon request.

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
