# Peer review of "Paramecium bursaria as a Potential Tool for Evaluation of Microplastics Toxicity"

_biology, 2022, doi:10.3390/biology11121852_

Round 1
Reviewer 1 Report
Remarks to the Author:
Jianhua Zhang and colleagues determined the feasibility of unicellular Paramecium bursaria as a potential in vivo model organism for toxicological assessment of microplastics by detecting the abnormal swimming patterns, moving speed, predator avoidance and oxidative stress. Most of the results are convincing evidences, but some are unconvincing.
Overall, I would support the publication of this study once the authors have addressed a series of changes.
My comments:
1. It is necessary to add some introduction on the relevant concentrations of microplastics in model organisms when they produce acute and chronic toxic effects.
2. Line 55: This sentence should be edited again. “However, elicited changes in response to MPs in unicellular animals (i.e., protozoan) and most importantly, potentials of protozoan as alternative model species for evaluating MPs toxicities are currently underappreciated.”
3. Line108 and line 124: “1 × 102 to 1 × 109 particles/mL” “1 × 108 particles/mL” Why did the author choose these concentrations? What is the basis for selection?
4. Scale bar should be added in the Figure 2.
5. In Figure 3. What is the possible reason for the difference between mean and max speed. And what is the significance of measuring the maximum speed
6. Line 258: The author mentioned aquatic species, but only plankton was mentioned in Reference [28]. More references about other aquatic species should be added after the sentence “Although toxicity of MPs have been tested in a range of aquatic species”. The author can also add some other model animals for comparison and discussion.
7. Line 330: Activities of two canonical antioxidant enzymes SOD and CAT were quantified in MPs-exposed P. bursaria. Have the authors tried to use Reactive Oxygen Species Assay Kit to detect the level of reactive oxygen species in P. bursaria? After all, the enzyme activities are not completely equal to the level of oxidative stress.
8. Line 512: “This observation is likely due to combinatorial effects caused by MPs exposure. Firstly, chemoreceptors responsible for detecting danger signals are disrupted, which render MPs-exposed P. bursaria less sensitive to toxic cues.” Can the author find some references to support the statement "chemoreceptors responsible for detecting danger signals are disrupted”?
Author Response
Thank you for taking the time to review this manuscirpt and please see the attachment of which is our point-by-point responses.

Reviewer 2 Report
The manuscript of Zhang et al “Paramecium bursaria as a potential in vivo model organism for toxicological assessment of microplastics” concerns an important problem of contamination of the environment with plastics. Although the topic seems important, and the results of the experiments, presented graphs and statistical analysis do not cause any questions, the study presented in the manuscript does not seem a toxicological study.
In the case of plastic microparticles it is not much the question of toxicity, but the problem of an extra load and the presence of extra hard particles in the cytoplasm, which the ciliate has to cope with. Since paramecia serve as a prey to crustaceans and other metazoans, they can be regarded as a vector to deliver the MPs to invertebrates and thus cause digestive problems in the predators, but again, this would not be due to toxicity of microplastics, but due to blocking of the digestive tract. The adverse effect of MPs on the ecosystem could be estimated by assessing the uptake of MPs in ciliates, but this effect is not caused by toxicity. If the authors wanted to check toxicity of plastic microparticles (in fact, they are dealing only with polystyrene, while there can be other types of plastic), they should have used one more control, for instance, loading the ciliates with some inert material, for instance, glass beads.
Besides that, in my opinion, compared to other ciliates, for instance, Loxodes, paramecia possess a rather robust pellicle and are rather tolerant to many chemicals, so, in general, I would not recommend using them in the toxicological tests. Thus, the title of the manuscript has to be changed so that it should correspond to the results obtained in the study. I would certainly avoid the words “toxicological assessment” and “model organism”.
Another point of concern is the observed effect of glycerol. Since the speed of ciliates treated with MPs is less than that of control cells, the latter are possibly quick enough to escape from medium containing the threshold limit concentration of glycerol, which affects the membrane properties and even may cause membrane permeabilization in the more slowly moving ciliates loaded with MPs. Membrane permeabilization in the ciliates loaded with MPs could cause changes in the potassium and calcium channels, which, in its turn, could change the ciliate behavior. I would suggest using some other repellent which would not cause quick changes in the membrane properties. Alternatively, I would recommend a more careful discussion of this experiment.
In line 383 the authors suggest that MPs could leak from the food vacuoles: “punctate staining … possibly represented the MPs leaked from the food vacuoles”
In my opinion, leakage of the digestive vacuoles presumes membrane permeabilization, which seems an unlikely event, or, possibly, the authors just use the wrong term to describe the phenomenon. It is more likely that some of MPs could use the route of endosymbiotic zoochlorellae and just pinch off the food vacuoles surrounded by a membrane derived from the digestive vacuole (see Kodama, Fujishima 2009). MPs, like zoochlorellae could be later translocated to the paramecium cortex and reside there among the trichocysts. This suggestion seems to be more plausible, however, one needs to perform electron microscopy to check this possibility. Anyway, this possibility should be discussed in the discussion section.
Line 396-398 “Although underlying mechanisms remain to be investigated, physical disruption of membrane integrity due to MPs exposure has been documented [38]. Consequently, it is possible that the locomotory organ of P. bursaria, i.e., cilia, was physically damaged by MPs and thus their locomotor activities were impaired”. I doubt very much that the cilia are damaged by MPs, since I can hardly imagine how they can get inside the cilia taking into account the mechanism of intraflagellar transport. Again, one needs electron microscopy to check if the MPs get inside the cilia. I would try to find some more plausible explanation for changing of locomotion.
So, discussion section should be revised.
Lines 62-63 : “Paramecium is a genus of ciliates with short hairs (i.e., cilia) arranged in rows on the membrane” Cilia are not hairs, it would be incorrect to put these two notions equal in a scientific journal
I would also advise the authors to perform a thorough check of the language. Below you can find some typos I have noticed:
Line 20: should be “which may serve”
Line 29: should be “showed” instead of “shown”
Line 31: please, delete the articles, should be “reduction”, “evaluation”
Line 54: should be “knowledge is”
Line 55: should be either “Metazoa” or “metazoans” (since you use plural)
Line 56: should be either “Protozoa” or “protozoans”
Line 57: should be “protozoans”
Line 59: should be “Protozoans”
Lines 60-61 should be “Paramecium is….and has been used as a valuable model organism”
Line 81: should be “showed” instead of “shown”, the same in lines 201, 206, 299
Line 84: did you mean “will serve”?
Line 117: please, put “fixed” instead of “fixated”
Line 139: I am afraid, there is no term “miller meter”, you must have meant “millimeter”
Line 196: should be “first”, not “firstly”
Line 323: should better be “which correlated”
Line 391 should be “are directly linked”
Line 394 should be “that took up”
Line 395 should be “accompanied by”
Author Response

(The authors gave the same response as above.)

Round 2
Reviewer 2 Report
All my concerns have been addressed, and the revised discussion section looks much more convincing. All language inaccuracies have been corrected. A couple of the remaining ones - lines 361-363. According to rules of scientific writing in English, the results obtained by the authours of the manuscript should be in Past Simple, and the results of other investigators - in Present Perfect. So, it would be better to put "our results showed" and "we did not intend to". In the rest, the manuscript can be published in the present form.